# Efficacy of Modified Treat-and-Extend Regimen of Aflibercept for Macular Edema from Branch Retinal Vein Occlusion: 2-Year Prospective Study Outcomes

**DOI:** 10.3390/jcm10143162

**Published:** 2021-07-17

**Authors:** Yusuke Arai, Hidenori Takahashi, Satoru Inoda, Shinichi Sakamoto, Xue Tan, Yuji Inoue, Satoko Tominaga, Hidetoshi Kawashima, Yasuo Yanagi

**Affiliations:** 1Department of Ophthalmology, Jichi Medical University, 3311-1 Yakushiji, Shimotsuke-shi, Tochigi 329-0498, Japan; r1003ya@jichi.ac.jp (Y.A.); r1208is@jichi.ac.jp (S.I.); r1136ss@jichi.ac.jp (S.S.); notani_s_0906@yahoo.co.jp (S.T.); hidemeak@khaki.plala.or.jp (H.K.); 2Japan Community Health Care Organization Tokyo Shinjuku Medical Center, 5-1 Tsukudocho, Shinjuku-ku, Tokyo 162-8543, Japan; tanxue1201@hotmail.com; 3Department of Ophthalmology, Graduate School of Medicine, University of Tokyo, 7-3-1 Hongo, Bunkyo-ku, Tokyo 113-8655, Japan; 4Department of Ophthalmology, Teikyo University, 2-11-1 Kaga, Itabashi-ku, Tokyo 173-8605, Japan; yino-tky@umin.ac.jp; 5The Ophthalmology & Visual Sciences Academic Clinical Program, Duke-NUS Medical School, National University of Singapore, Singapore 169857, Singapore; yanagi.yasuo@icloud.com

**Keywords:** branch retinal vein occlusion, anti-vascular endothelial growth factor, modified treat-and-extend regimen, prospective multicenter intervention study

## Abstract

This study aimed to evaluate the long-term (24-month) efficacy and safety of a modified treat-and-extend (mTAE) regimen of aflibercept for macular edema (ME) due to branch retinal vein occlusion (BRVO). This was a prospective multicenter intervention study. We evaluated 50 eyes in 50 patients with ME due to BRVO enrolled between October 2016 and September 2017. The patients received intravitreal aflibercept (IVA) injections according to a mTAE regimen for 24 months. This study reports the secondary endpoints of best-corrected visual acuity (BCVA) and central subfield thickness (CST) at 24 months and compares them with previously reported primary endpoints. Compared with baseline BCVA and CST of 0.33 (0.27) and 488 (165) µm (mean (standard deviation)), respectively, BCVA and CST were significantly improved at 12 and 24 months (12 months: 0.059 (0.19) LogMAR and 299 (112) µm; 24 months: 0.034 (0.18) LogMAR and 272 (81) µm, respectively; both *p* < 0.0001). Over the 24-month period, the mean number of IVA injections and clinic visits was 7.4 (3.3) and 11.1 (2.0), respectively. The mTAE regimen of IVA injections for ME due to BRVO was effective for improving BCVA and reducing CST over 24 months. This regimen shows promise for reducing the number of injections and clinic visits.

## 1. Introduction

Branch retinal vein occlusion (BRVO) is one of the most common retinal vascular diseases. The prevalence of BRVO reported by studies conducted in the United States, Europe, Asia, and Australia is 0.4%, whereas in Japan it is 2.0% [1,2]. Anti-vascular endothelial growth factor (VEGF) therapy is currently the standard treatment for macular edema (ME), secondary to BRVO [3]. Dexamethasone intravitreal implant has been reported as another treatment option [4]. Many studies have demonstrated the efficacy of anti-VEGF for BRVO [5,6,7,8,9], but as many as 50% of patients still required intravitreal ranibizumab (IVR) injections 4 years after the first treatment [9]. We therefore consider treatment of chronic BRVO to be burdensome for both patients and healthcare workers. In addition, real-world data shows that treatment outcomes in actual practice were not as impressive as the seminal randomized controlled studies, possibly due to undertreatment [10]. In previous reports, most studies used a pro re nata (PRN) regimen, whereas some studies supported the use of a treat-and-extend (TAE) regimen; however, a more optimal regimen that avoids undertreatment and reduces the number of intravitreal injections and clinic visits has been sought.

To this end, we developed a modified treat-and extend (mTAE) regimen of intravitreal aflibercept (IVA) injections for ME due to BRVO and initiated a prospective multicenter intervention study to assess the effectiveness of the regimen. We previously reported the primary outcomes of best-corrected visual acuity (BCVA) and reduced central subfield thickness (CST) at 12 months after the initial injection and found that the mTAE regimen was effective and required fewer injections and clinic visits [11].

Here, to evaluate whether the efficacy and safety of the regimen is maintained and whether the reduced burden on patients and healthcare staff continues over the long term, we report the 24-month results of this mTAE regimen.

## 2. Material and Methods

We have described the materials and methods in detail in our previous report on outcomes at 12 months [11] and provide a brief summary here.

### 2.1. Study Design

This prospective multicenter intervention study involving patients with treatment-naïve ME due to BRVO commenced in October 2016 with the last patient completing the 1-year study in September 2018.

The study protocol was approved by the institutional review board of Jichi Medical University (B18-003). The study followed the tenets laid out in the Declaration of Helsinki. Written informed consent was obtained from all patients. The study was registered in the University Hospital Medical Information Network Clinical Trials registry prior to study commencement (27/10/2016, UMIN000024587).

### 2.2. Patients and Ophthalmological Examination

This study was conducted at 6 sites in Japan (Institution A, Jichi Medical University Hospital; B, Japan Community Health Care Organization Tokyo Shinjuku Medical Center; C, Ohkubo Eye Clinic; D, Takahashi Eye Clinic; E, Saito Eye Clinic; and F, Aoki Eye Clinic). The detailed inclusion and exclusion criteria are described in our previous report [10]. Briefly, the study included treatment-naïve patients aged ≥20 and <90 years with visual impairment due to ME secondary to BRVO who underwent a complete ophthalmic examination.

### 2.3. mTAE Regimen

Patients were treated with aflibercept according to the mTAE regimen we described in detail previously [11]. Appendix A show an example of the administration methods of PRN and TAE regimens compared with our mTAE regimen. Briefly, after the first injection, all patients attended follow-up visits every 4 weeks. A loading dose was not administered. If there was any exudative lesion in the macula (ME and/or serous retinal detachment), the patient received a second injection and the TAE process commenced. At that time, no CST criteria were applied. After the second injection, if there was no exudative lesion, the patient received an IVA injection and the period to the next treatment was extended by 4 weeks at a time (no maximum interval was specified). This interval of 4 weeks was selected instead of 2 weeks because the degree of ME secondary to BRVO was considered to be not so severe, and 4 weeks was considered adequate to prevent retinal damage. If there was increased exudative change in optical coherence tomography findings at first recurrence, the patient received an injection and the visit interval was shortened by 4 weeks; if the exudative change was the same or less compared with the first recurrence, an injection was given and the interval to the next visit remained unchanged. Patients who maintained a dry macula after the first injection were followed up every 4 weeks until week 16, and thereafter, the visit interval could be extended to 3 months. If the first recurrence of exudative changes was observed after week 16, the TAE regimen was recommended at 3-month intervals. Patients who had no recurrence after week 16 were observed every 3 months.

### 2.4. Outcome Measures

The main outcome measures were mean changes in BCVA and CST, with the primary outcomes evaluated at 12 months [11] and the secondary outcomes evaluated at 24 months. We also examined the numbers of IVA injections and clinic visits over the 24-month period, as well as the distribution of IVA treatment intervals. In addition, we assessed the changes in BCVA and CST compared with baseline and 12-month values as well as calculated the numbers of IVA injections and clinic visits according to type of BRVO: macular BRVO (i.e., occlusion of only the vein inside the arcade vessels) and major BRVO (all BRVO with ME other than macular BRVO). Major BRVO is caused by occlusion of one of the four major branch retinal veins, and it involves the entire segment of the retina drained by the vein extending all the way to the peripheral retina. Macular BRVO is caused by occlusion of one of the veins from only the macular region (the part of the retina between the superior and inferior vascular arcades) [12].

### 2.5. Statistical Analysis

As in our evaluation of the primary outcomes [11], statistical analysis was performed using JMP Pro software version 14.1.0 (SAS Institute, Cary, NC, USA). The efficacy endpoints were analyzed in the full analysis set (FAS), which included all patients who received any study treatment and had data at baseline and at least 1 time point after baseline. Between 4 and 23 months, the last observation carried forward was used to impute the missing data. Major BRVO and macular BRVO were compared in 39 of 50 cases with per protocol set (PPS), excluding 11 that dropped out. Decimal BCVA was converted to logMAR. BCVA and CST were compared with those at the first injection by using a two-tailed paired *t*-test. 

## 3. Results

### 3.1. Baseline Characteristics

Table 1 shows the baseline characteristics of the participants. As reported previously [11], among the 50 patients included in this study (24 men, 26 women), 11 dropped out (8 were lost to follow-up, 1 withdrew consent, 1 underwent cataract surgery, and 1 underwent pan-retinal photocoagulation). In total, 46 eyes were observed for 12 months, and 39 eyes were observed for 24 months. Mean age (SD) at the start of treatment was 66 (12) years (range 42–85 years). Mean baseline BCVA (logMAR) was 0.33 (0.27) and mean baseline CST was 488 (165) µm. Major BRVO was evident in 25 eyes and macular BRVO in 14 eyes. Figure 1 showed the study design and patient disposition.

### 3.2. BCVA and CST Outcomes

Mean BCVA was significantly improved at 1, 12, and 24 months (0.11 (0.16), 0.059 (0.18), and 0.034 (0.18), respectively) compared with baseline (0.33 (0.27); all *p* < 0.0001; Figure 2).

In three eyes, the final BCVA was less than logMAR0.3: in two of these eyes, the BCVA was equivalent to that at the first visit, and in one eye, it declined due to cataract progression. The mean CST was significantly decreased after IVA injections at 1, 12, and 24 months (248 (36) µm, 299 (112) µm, and 272 (81) µm, respectively) compared with baseline (480 (165) µm; all *p* < 0.0001; Figure 3).

### 3.3. IVA Injections and Clinic Visits

The mean number of IVA injections over the 24 months was 7.4 (3.3), with significantly fewer injections administered in the second year compared with the first year (2.9 (1.4) vs. 4.7 (2.0); *p* = 0.024, paired *t*-test). Five eyes received only one injection (Figure 4). 

The mean number of clinic visits was 11.1 (2.0) over the 24 months, with significantly fewer visits occurring in the second year than in the first year (3.9 (1.4) vs. 7.1 (1.0); *p* < 0.0001, paired *t*-test). (Figure 5).

### 3.4. Time to First Recurrence from First Injection

Recurrence occurred most frequently at 3 months (15 eyes). Recurrence of exudative changes was apparent in eight eyes at 1 month and in two eyes at 7 months. These two eyes showed no recurrence of exudative change after the TAE regimen was started. Five eyes showed no recurrence over the 24 months, three with major BRVO, and two with macular BRVO.

### 3.5. Injection Frequency

The IVA injection interval could be extended to more than 5 months in 21 eyes (54%), including the 5 eyes that showed no recurrence over the 24 months (Figure 6).

### 3.6. Major and Macular BRVO

Table 2 shows the comparison between major BRVO and macular BRVO.

Figure 7 shows the improvements in BCVA and CST found in 25 eyes with major BRVO and 14 eyes with macular BRVO.

Mean BCVA in major and macular BRVO was significantly improved at 12 and 24 months after the initial injection (12 months: 0.058 (0.18) and 0.037 (0.20); 24 months: 0.061 (0.20) and 0.029 (0.15), respectively) compared with baseline (0.32 (0.25) and 0.34 (0.30), respectively; both *p* < 0.0001, paired *t*-test). Mean CST in major and macular BRVO was significantly decreased at 12 and 24 months (12 months: 299 (135) µm and 258 (52) µm; 24 months: 270 (38) µm and 259 (90) µm, respectively) compared with baseline (494 (168) µm and 454 (141) µm, respectively; both *p* < 0.0001, paired *t*-test).

There was no significant difference in either BCVA or CST at 24 months between major and macular BRVO (*p* = 0.84 and *p* = 0.53, respectively; paired *t*-test). There was also no significant difference between major and macular BRVO in the mean number of IVA injections administered (7.5 (3.6) and 7.1 (3.0), respectively; *p* = 0.69, paired *t*-test) or in the mean number of clinic visits required (11.3 (2.2) and 10.6 (1.6), respectively; *p* = 0.31, paired *t*-test). The mean time from awareness of symptoms to diagnosis was significantly shorter in major BRVO than in macular BRVO (1.5 (1.6) and 3.3 (3.8), respectively; *p* = 0.041, paired *t*-test).

### 3.7. Safety over the 24 Months

No serious ocular complications associated with IVA injections were observed. Cataract progressed in only one eye and was not a complication of the IVA injection. BCVA was improved in the eye after cataract surgery. No serious systemic adverse events were noted in any eyes over the 24 months.

## 4. Discussion

In this study, we demonstrated the long-term effectiveness of the mTAE regimen of IVA injections for improving VA and reducing CST in ME due to BRVO. To our knowledge, this is the first prospective study of an mTAE regimen using aflibercept that combines the best of the PRN and TAE regimens and reports long-term outcomes. Considering the relatively good baseline VA in the present study (0.33 LogMAR unit, which is equivalent to 68.5 ETDRS letters), we believe that the improvement in BCVA and the reduction in CST values are comparable with those in previous studies, such as the VIBRANT study, a randomized controlled trial conducted using aflibercept in a fixed protocol for regulatory approval [13], and the BRIGHTER study, which confirmed the efficacy of less treatment-intensive treatment regimen using IVR injections, that is, one injection plus PRN [14]. Over 2 years, our mTAE regimen maintained BCVA and CST with fewer IVA injections compared with the PRN regimen of IVR injections used in the BRIGHTER study (7.4 (3.34) vs. 11.4 (5.81) injections, respectively). Moreover, our regimen also resulted in fewer clinic visits (11.1 (2.0) vs. 24, respectively). In the second year of our regimen, there were fewer IVA injections and clinic visits compared with the first year (2.9 injections and 3.9 visits). In previous studies, the number of injections after the second year also decreased compared with the first year [9,15]. Analysis of real-world data of 2530 eyes from 48 real-world studies demonstrated that the mean baseline VA in the pooled data was 54.0 letters and the change in VA at 12 months was 14.6 letters, with a change in CST of -181.7 mm, suggesting that visual and anatomical gains were not as impressive as those in the seminal randomized control study [10]. This is, at least in part, due to reduced injection frequency. To address such real-world problems, in the field of exudative age-related macular degeneration, a TAE approach, rather than a PRN approach, is recommended, considering it in terms of predictability and lower treatment burden, which will improve service capacity [16]. We believe such advantages of TAE rationalize its use for the treatment of ME due to BRVO as well. This is especially relevant for countries such as Japan given that Ogura et al. reported that most retina specialists adopt PRN regimens regardless of ME severity [17].

We believe that the “modified” TAE regimen in this study contributes to avoiding overtreatment. Specifically, in this study, the time of the first recurrence varied, highlighting the need to individualize the administration method for each patient. Five eyes showed no recurrence at all. As such, we believe that choosing a “standard” TAE regimen from the beginning might have led to overtreatment in such patients, and our mTAE regimen may prevent this from occurring. Importantly, the IVA injection interval could be extended to more than 5 months in 21 eyes (54%) in this study (including in the 5 eyes that showed no recurrence). Our results may also indicate that IVA injections can be stopped if the injection interval can be extended to 4 months because patients in this study for whom the injection interval could be extended to 4 months or longer did not have recurrence or had only small exudative changes over the entire observation period. For patients who require frequent injections, it is important that the administration regimen and the interval between hospital visits place as low a burden as possible. In this aspect, the injection interval could not be extended beyond 3 months in nine eyes (23.1%). In the RETAIN study, 50% of patients still required IVR injections 4 years after the first treatment [9]. Thus, although anti-VEGF therapy can maintain good VA, roughly one-third to half of the patients need injections over the long term. Additionally, chronic BRVO may require additional treatment such as steroids and laser photocoagulation [18,19], further highlighting the importance for retina specialists to secure time for patient management by reducing outpatient visits. In this study, five cases that showed no recurrence were followed up every 3 months after month 4, so the number of clinic visits increased to 11 times. For example, if the first recurrence occurred in month 4 and no recurrence thereafter, the administration period was steadily extended because there is no upper limit on the administration period. As a result, the number of clinic visits was nine. The number of clinic visits would be further reduced if the follow-up period for one-injection cases can be extended.

Based on previous studies, it is generally assumed that major BRVO is more aggressive than macular BRVO. A previous study found that macular BRVO responded better than major BRVO to treatment, with clearer improvement of vision and fewer injections [20]. Similar results are reported for conbercept, which had a better short-term effect in patients with macular BRVO than in those with major BRVO [21]. In the present study, however, neither the change in BCVA nor CST over 24 months was significantly different between major and macular BRVO. Only the mean time from awareness of symptoms to diagnosis was significantly shorter in major BRVO, which may reflect patients being more likely to notice symptoms when lesions are large. Our study suggests that aflibercept is equally effective for both major and macular BRVO; however, further studies are warranted to confirm whether the treatment efficacy of aflibercept is equivalent for both major and macular BRVO.

There are some limitations to this study. First, the study was designed as a single-arm trial and the sample size was relatively small. Second, only Japanese patients were included. Therefore, further prospective studies with larger study populations and longer follow-up are needed to confirm the usefulness of our regimen. 

## 5. Conclusions

In this prospective study, an mTAE regimen of IVA injections for treating ME due to BRVO effectively improved BCVA and reduced CST over a 2-year period. We believe that this mTAE regimen of aflibercept may be useful in actual clinical practice, by improving VA and structural outcomes as well as reducing the number of required injections and hospital visits, thereby also helping to reduce burden on both patients and healthcare workers.

## Figures and Tables

**Figure 1 jcm-10-03162-f001:**
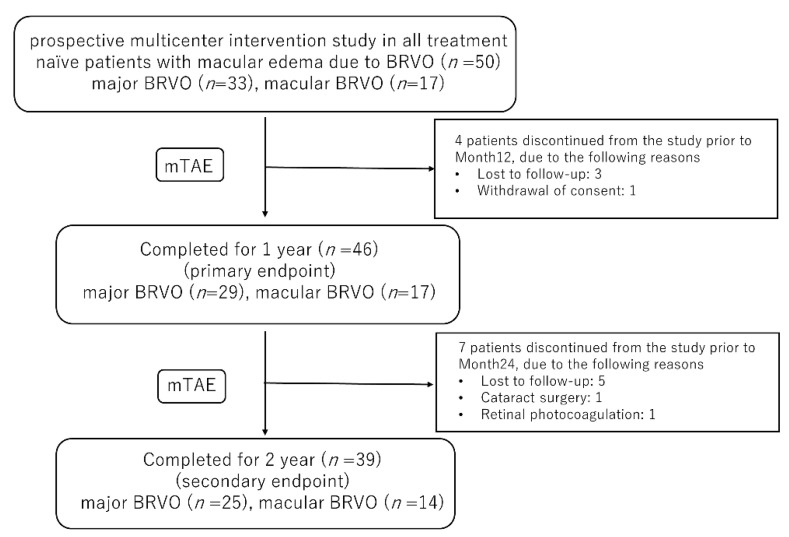
The study design and patient disposition. BRVO, branch retinal vein occlusion, mTAE, modified treat-and-extend.

**Figure 2 jcm-10-03162-f002:**
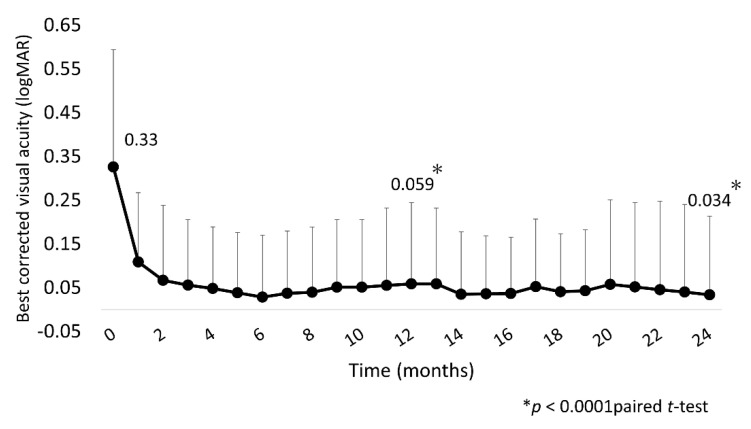
Mean BCVA (logMAR) over 24 months. Mean BCVA improved significantly at 1 month after IVA injection, with the improvement continuing through to 24 months. BCVA, best-corrected visual acuity; IVA, intravitreal aflibercept.

**Figure 3 jcm-10-03162-f003:**
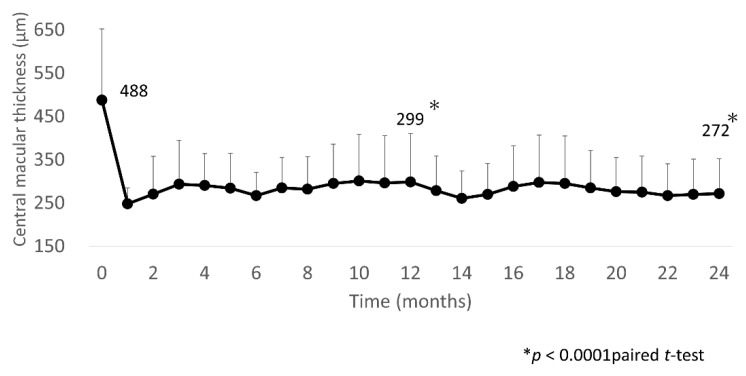
Mean CST over 24 months. Mean CST improved significantly at 1 month after IVA injection, with the improvement sustained through to 24 months. CST, central subfield thickness; IVA, intravitreal aflibercept.

**Figure 4 jcm-10-03162-f004:**
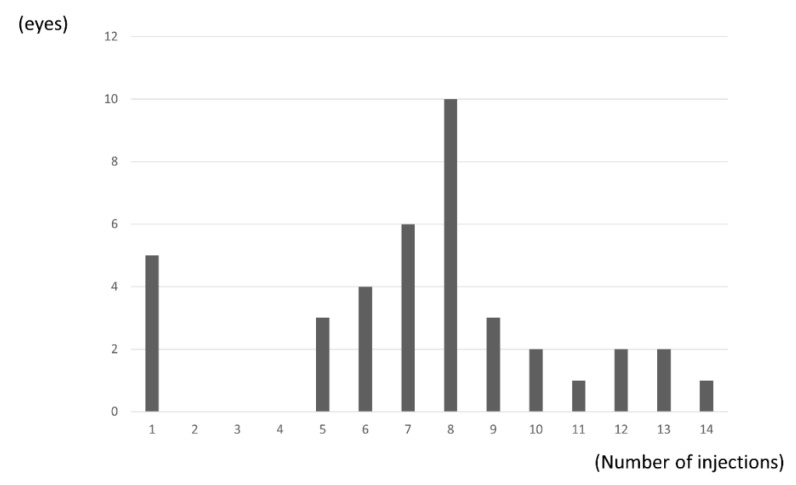
Distribution of number of injections. Five eyes (13%) received only one injection. Mean number of IVA injections over the 24 months was 7.4 (3.3).

**Figure 5 jcm-10-03162-f005:**
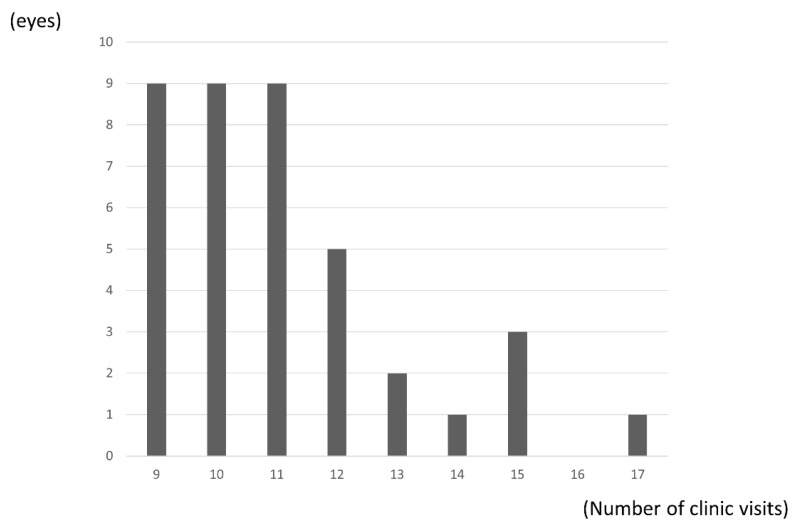
Distribution of number of clinic visits. Mean number of clinic visits was 10.1 (2.0) over the 24 months.

**Figure 6 jcm-10-03162-f006:**
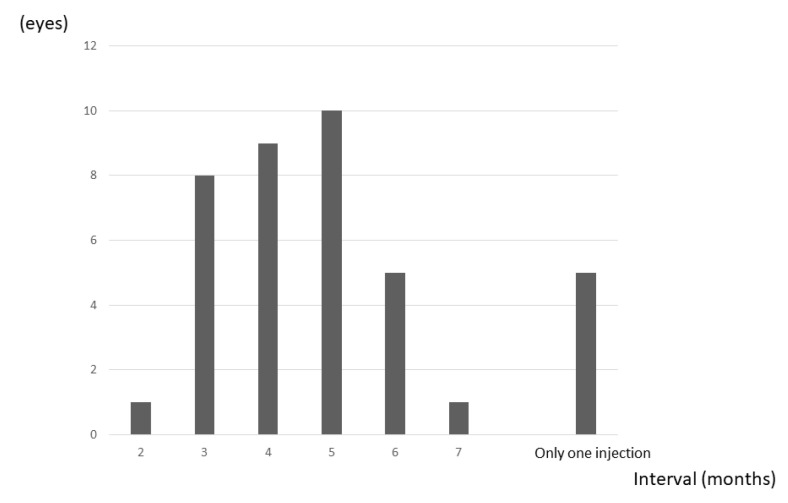
Frequency of maximum dosing interval. Five eyes (13%) had no recurrence. The IVA injection interval could be extended to more than 5 months in 21 eyes (54%) (including all 5 eyes showing no recurrence).

**Figure 7 jcm-10-03162-f007:**
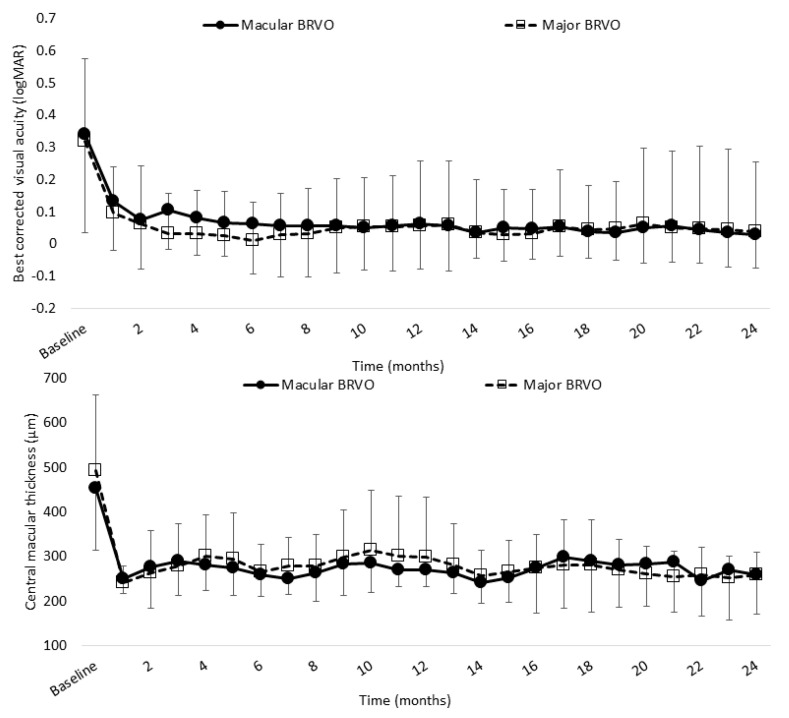
Mean BCVA (logMAR) and CST of major BRVO and macular BRVO over 24 months. Mean BCVA in major and macular BRVO was significantly improved at 12 and 24 months (0.085 (0.20) and 0.061 (0.23), 0.0017 (0.14) and −0.015 (0.11), respectively) compared with baseline (0.33 (0.26) and 0.30 (0.30), respectively; both *p* < 0.0001, paired *t*-test). Mean CST in major and macular BRVO was significantly decreased at 12 and 24 months (299 (135) µm and 257 (52) µm, (270 (129) µm and 259 (70) µm, respectively) compared with baseline (494 (168) µm and 454 (141) µm, respectively; both *p* < 0.0001, paired *t*-test). BCVA, best-corrected visual acuity; BRVO, branch retinal vein occlusion; CST, central subfield thickness.

**Table 1 jcm-10-03162-t001:** Baseline characteristics.

Cases	50
Age (years; mean ± SD)	66 ± 12
Sex (male/female)	24/26
BCVA (logMAR; mean ± SD)	0.33 ± 0.27
CMT (µm; mean ± SD)	488 ± 171

BCVA, best-corrected visual acuity; CMT, central macular thickness; SD, standard error.

**Table 2 jcm-10-03162-t002:** Comparison of major BRVO and macular BRVO.

	Major BRVO (*n* = 25)	Macular BRVO (*n* = 14)	*p*-Value *
Baseline BCVA (logMAR; mean ± SD )	0.33 ± 0.26	0.30 ± 0.30	0.68
Baseline CRT (µm; mean ± SD)	509 ± 161	477 ± 191	0.28
BCVA at month 24 (logMAR; mean ± SD)	0.0017 ± 0.14	−0.026 ± 0.074	0.15
CRT at month 24 (µm; mean ± SD)	257 ± 52	259 ± 70	0.94
Mean number of injections (times; mean ± SD)	7.52 ± 3.57	7.07 ± 2.97	0.69
Mean number of visits (times; mean ± SD)	11.32 ± 2.19	10.64 ± 1.55	0.31
Duration between symptoms and initial therapy (month; mean ± SD)	1.46 ± 1.55	3.26 ± 3.76	0.041

* paired *t*-test. SD, standard error.

## Data Availability

All data generated or analyzed during this study are included in this published article and its Appendix A. All data set files are available from the Figshare database (https://doi.org/10.6084/m9.figshare.8982500, accessed on 6 May 2021).

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
