# Peer review of "Efficacy of Modified Treat-and-Extend Regimen of Aflibercept for Macular Edema from Branch Retinal Vein Occlusion: 2-Year Prospective Study Outcomes"

_jcm, 2021, doi:10.3390/jcm10143162_

Round 1

Reviewer 1 Report

This is a paper that studies the T&E modified treatment in BRVO ou macular occlusion. This is an interesting topic that compares with PRN treatment studies. This is very well structured and very well written. Main objective and secondary objectives are clear and well specified. 

Author Response

Response to Reviewer 1 Comments

Point1. This is a paper that studies the T&E modified treatment in BRVO ou macular occlusion. This is an interesting topic that compares with PRN treatment studies. This is very well structured and very well written. Main objective and secondary objectives are clear and well specified. 

Thank you for your helpful comment. I have consulted the English editing service again in full text.

Reviewer 2 Report

Introduction: 1. Not only the AntiVEGF is the treatment Option. There is also Dexamethasone Implant.  Why Ranibizumab is mentioned only. 

 Material and Metods:  Please describe mTAE in detail in this paper too, otherwise person reading this paper has to read your first paper. However then this paper is not complete.

How do you calculated the 'n' ? 

Results: n=25  major BRVO and n=15 macular BRVO and  n=10 patients are not mentioned.  

Why 4 weeks TAE interval ? why not 2 weeks ?

Please describe your Study protocol in a Figure and explain.

line 142, 169 Font is smaller Please check the whole paper and correct it as needed !

Author Response

Response to Reviewer 2 Comments

Point1. Not only the AntiVEGF is the treatment Option. There is also Dexamethasone Implant.  Why Ranibizumab is mentioned only. 

Thank you for your comment. We added about the dexamethasone implant in the manuscript. (line -)

Point2. Please describe mTAE in detail in this paper too, otherwise person reading this paper has to read your first paper. However then this paper is not complete.

Thank you for your comment. We added about the explanation about mTAE rejimen in the manuscript. (line-)

Point3. How do you calculated the 'n' ? Results: n=25  major BRVO and n=15 macular BRVO and  n=10 patients are not mentioned.  

Thank you for your comment. The analysis of major BRVO and macular BRVO was performed except for 11 cases that dropped out from 50 cases. We added about this in the manuscript. (line)

Point4. Why 4 weeks TAE interval ? why not 2 weeks ?

Thank you for your comment. We thought that the degree of ME of BRVO was not so strong and that retinal damage could be sufficiently prevented at intervals of 4 weeks. We added about this in the manuscript. (line)

Point5. Please describe your Study protocol in a Figure and explain

Thank you for your comment. We describe our study protocol in a Figure1.

Point5. line 142, 169 Font is smaller Please check the whole paper and correct

it as needed !

Thank you for your comment. We Corrected the font of the whole sentence.

We have also revised the English in the manuscript and added some references.
